# The prevalence of symptoms in 24,410 adults infected by the novel coronavirus (SARS-CoV-2; COVID-19): A systematic review and meta-analysis of 148 studies from 9 countries

**Michael C. Grant**[1], **Luke Geoghegan**[2], **Marc Arbyn**[3], **Zakaria Mohammed**[4,5], **Luke McGuinness**[6,7], **Emily L. Clarke**[8,9], **Ryckie G. Wade**[4,5]*

1 University of Sheffield, Broomhall, Sheffield, United Kingdom, 2 Academic section of vascular surgery, department of surgery and cancer, Imperial College London, London, United Kingdom, 3 Unit Cancer Epidemiology–Belgian Cancer Centre, Sciensano, Belgium, 4 Leeds Institute for Medical Research, University of Leeds, Leeds, United Kingdom, 5 Department of Plastic and Reconstructive Surgery, Leeds Teaching Hospitals, Leeds, United Kingdom, 6 Population Health Sciences, Bristol Medical School, University of Bristol, Bristol, United Kingdom, 7 MRC Integrative Epidemiology Unit at the University of Bristol, Bristol, United Kingdom, 8 Division of Pathology and Data Analytics, University of Leeds, Leeds, United Kingdom, 9 Department of Histopathology, Leeds Teaching Hospitals, Leeds, United Kingdom

* ryckiewade@gmail.com

**Data Availability Statement:** The raw extracted data and additional relevant files are available via an

## Abstract

### Background

To limit the spread of SARS-CoV-2, an evidence-based understanding of the symptoms is critical to inform guidelines for quarantining and testing. The most common features are purported to be fever and a new persistent cough, although the global prevalence of these symptoms remains unclear. The aim of this systematic review is to determine the prevalence of symptoms associated with COVID-19 worldwide.

### Methods

We searched PubMed, Embase, CINAHL, AMED, medRxiv and bioRxiv on 5th April 2020 for studies of adults (>16 years) with laboratory test confirmed COVID-19. No language or publication status restrictions were applied. Data were independently extracted by two review authors into standardised forms. All datapoints were independently checked by three other review authors. A random-effects model for pooling of binomial data was applied to estimate the prevalence of symptoms, subgrouping estimates by country. $I^2$ was used to assess inter-study heterogeneity.

### Results

Of 851 unique citations, 148 articles were included which comprised 24,410 adults with confirmed COVID-19 from 9 countries. The most prevalent symptoms were fever (78% [95% CI 75%-81%]; 138 studies, 21,701 patients; $I^2$ 94%), a cough (57% [95% CI 54%-60%]; 138 studies, 21,682 patients; $I^2$ 94%) and fatigue (31% [95% CI 27%-35%]; 78 studies, 13,385 patients; $I^2$ 95%). Overall, 19% of hospitalised patients required non-invasive ventilation (44 studies,

Open Science Framework repository (https://doi.org/10.17605/OSF.IO/9VX6U).

**Funding:** Ryckie G Wade is a Doctoral Research Fellow funded by the National Institute for Health Research (NIHR, DRF-2018-11-ST2-028). Luke McGuinness is supported by an NIHR Doctoral Research Fellowship (DRF-2018-11-ST2-048). The views expressed are those of the author(s) and not necessarily those of the NHS, the NIHR or the Department of Health and Social Care. Marc Arbyn was supported by the VALCOR project (Sciensano, Brussels, Begium). The funders had no role in the initiation, conduct or publication of this project.

**Competing interests:** The authors have declared that no competing interests exist.

6,513 patients), 17% required intensive care (33 studies, 7504 patients), 9% required invasive ventilation (45 studies, 6933 patients) and 2% required extra-corporeal membrane oxygenation (12 studies, 1,486 patients). The mortality rate was 7% (73 studies, 10,402 patients).

## Conclusions

We confirm that fever and cough are the most prevalent symptoms of adults infected by SARS-CoV-2. However, there is a large proportion of infected adults which symptoms-alone do not identify.

## Introduction

The novel coronavirus (SARS-CoV-2; 2019-nCoV; COVID-19) pandemic is a global crisis. As of April 10th 2020, there were over 1.5 million confirmed cases of whom over 92,000 have died [1]. In the absence of a vaccine or treatment with proven efficacy, limiting human-to-human transmission is critical [2–4]. Self-isolation (or self-quarantine) is an effective global strategy for limiting transmission following the emergence of symptoms [5] and equally, the manifestation of symptoms is used to guide testing.

Coronavirus is most infectious in the early phase of the illness [6,7] [8], so screening people with compatible symptoms [9] is fundamental to determining who should be quarantined and be tested [9]. Several systematic reviews have considered the symptoms of COVID-19 (amongst other parameters) [8,10–14] although all of them have major limitations. None systemically searched the grey literature (e.g. preprint archives such as medRxiv and bioRxiv) and in the context of a pandemic, the quality and quantity of the literature is evolving at speed [15]. Without incorporating all relevant preprints the findings of any systemic review will be weeks-months out-of-date at the time of publication [16]. Furthermore, with few included studies (30 in the largest and most recent [12]), the range of symptoms were limited and the estimates of prevalence are likely to be upwardly biased because only unwell patients (largely those admitted to hospital) were tested in the early phase of the outbreak.

To facilitate the rapid dissemination of high-quality open-science, there has been a surge of preprints related to COVID-19 manuscripts uploaded to the online archives medRxiv and bioRxiv [15]. The necessity to address deficiencies in current literature and potential to substantially improve the precision of estimates of symptom prevalence using both indexed and (the more voluminous and up-to-date) preprint literature from multiple geographical regions, represents the rational for this review.

The aim of this systematic review is to determine the prevalence of symptoms associated with COVID-19 worldwide.

## Methods

This systematic review was designed and conducted in accordance with the Cochrane Handbook of Systematic Reviews [17] and a pre-published protocol [18], and is reported in line with the Preferred Reporting Items for Systematic Reviews and Meta-Analyses (PRISMA) Statement (see S1 Checklist) [18,19].

### Participants and studies

Studies reporting the prevalence of patient-reported symptoms or clinician observed features in adults (>16 years) with laboratory confirmed novel coronavirus (SARS-CoV-2; covid-19)

derived from oro- or naso-pharyngeal swabs. We excluded case reports, articles which failed to disaggregate symptoms in adult and paediatric cohorts, studies of patients with prior respiratory infections (e.g. tuberculosis) or co-infections with other viruses (e.g. similar viruses SARS-CoV-1 or HCoV-EMC/2012, etc) and articles which we are unable to translate to English in a timely fashion.

## Target condition

The incubation period of COVID-19 is typically 5 days, but reported to last a maximum of 7 days [20–24]. The illness typically lasts 8 days [21]. Therefore, we will include any symptom(s) described up to 15 days before laboratory confirmed infection and during the illness.

## Search strategy

PubMed, Embase, AMED and CINAHL, medRxiv [25] and bioRxiv were interrogated according to our search strategy (S1 Appendix). Searches were limited to 1st January onwards. No language restrictions were applied.

## Study selection process

After de-duplication, all unique citations were independently screened by three review authors (MG, LG and RGW). The full texts of all potentially relevant articles were obtained. The reference lists for included articles and other systematic reviews were also scrutinised. Final lists of included articles were compared and disagreements resolved by consensus discussion between five authors (MG, LG, ZK, ELC and RGW).

## Data extraction

Two authors (MG and LG) independently extracted data and three authors (ZM, ELC and RGW) checked the accuracy of the extracted data using a standardised spreadsheet. Disagreements were resolved by discussion. We combined the following symptoms: "chest tightness" into the more prevalent symptom of wheeze; "shivers" and "chills" into rigors; malaise and "generalised weakness" (in the absence of any objective neurological deficit) into the more widely reported symptom of fatigue; conjunctivitis, conjunctival congestion and conjunctivital secretions into conjunctivitis. Where studies reported one symptom "or" another (e.g. nausea or vomiting) we did not extract this information as it was impossible to disaggregate. Where studies reported one symptom "and" another (e.g. nausea and vomiting) we extracted the prevalence of both. When studies grouped symptoms together (e.g. "respiratory symptoms") without further description or definition we were not able to extract this information.

## Methodological quality assessment

The risk of bias for included studies was not assessed for two main reasons: firstly, there is no consensus on ideal tool, nor one designed specifically for studies of prevalence [26] and secondly, such assessments would not change the approach to the modelling or presentation of the data, as per our protocol. Given such assessments are also time-intensive, we have taken the pragmatic decision to not perform risk of bias assessments.

## Analysis

The pooled prevalence of symptoms were estimated using the *metaprop* package [27] in Stata/MP v15. Dersimonian and Laird random-effects were used given the geographical and study-level heterogeneity. A random-effects model including Freeman-Tukey arcsine transformation

of the prevalence was used to normalise variance. 95% confidence intervals (CIs) were computed around the study-specific and pooled prevalence based on the score-test statistic. [28]. The variation in prevalence by country was assessed by subgroup meta-analyses and meta-regression. Statistical heterogeneity is assessed by $I^2$ which corresponds with the proportion of total variation due to inter-study heterogeneity and by p-values for inter-study heterogeneity within countries, between countries and overall [29]. Given the use of a random-effects model, inter-study heterogeneity within countries was only assessable when at least three studies were available. A z-test (and the corresponding p-values) assessed whether the observed prevalence was different from zero percent. Publication bias was not assessed.

## Results

### Study selection

Our search returned 2403 hits in PubMed, 2234 in Embase, 310 in CINAHL, 1 in AMED, and 434 preprints in medRxiv and bioRxiv on 5th April 2020. Following deduplication, the titles and abstracts of 851 unique records were assessed against the inclusion criteria. 743 of these were deemed to be potentially eligble. Full text screening then resulted in 148 included articles (Fig 1).

### Study characteristics

This review describes 24,410 adults with laboratory confirmed COVID-19 from 9 countries, including China [30,31,32,33,34,35,36,37,38,39,40,41,42,43,44–49,50–59,60–69,70–79,80–89,90–99,100–109,110–119,120–129,130–139,140–149,150–159,160–165], the UK [166,167], the USA [168,169], Singapore [170,171], Italy [172,173], Australia [174], Japan [175], Korea [176] and the Netherlands [177]. The mean age of patients was 49 years (SD 11) and where sex data were available, the ratio of males:females was 1.2:1 (10,306:8593). The characteristics of the included studies are shown in S1 Table.

Thirty-four studies reported that 845 of 7519 patients required non-invasive ventilation (pooled prevalence 17% [95% CI 11%-24%]; $I^2$ 98%). Forty-four studies (6513 patients) reported that 970 patients were admitted to an intensive care unit (pooled prevalence 19% [95% CI 13%-26%]; $I^2$ 97%). Forty-five studies (6933 patients) reported that 495 required invasive mechanical ventilation (pooled prevalence 9% [95% CI 6%-13%]; $I^2$ 95%). Twelve studies (1486 patients) reported that 2% of patients (36) required extra-corporeal membrane oxygenation (95% CI 0%-5%; $I^2$ 95%). Of the 73 studies that reported survival in 10,402 patients, there were 938 deaths (pooled prevalence 7% [95% CI 4%-11%]; $I^2$ 98%) which were attributable to COVID-19.

### Evidence synthesis

Table 1 shows the meta-analysed prevalence of symptoms, group by bodily system, and S2 Table shows the meta-analytical prevalence estimates from studies grouped by geographical region. The most prevalent symptom in patients with laboratory confirmed COVID-19 was a fever, experienced by 78% of patients (99% CI 75%-81%; Fig 2 and S2–S5 Figs). Whilst, there was substantial heterogeneity between countries ($I^2$ 94%) with estimates ranging from 83% in Singapore (99% CI 61%-98%) to 32% in Korea (99% CI 15%-51%), there was no evidence of a statistically significant difference between countries (S2 Table). A cough was the second most prevalent symptom, reported by 57% of test-positive patients (95% CI 54%-60%; Fig 3 and S6–S10 Figs). Whilst there was substantial heterogeneity between countries ($I^2$ 94%) with estimates ranging from 18% in Korea (99% CI 8%-36%) to 76% in the Netherlands (95% CI 66%-83%), there was no evidence of a statistically significant difference.

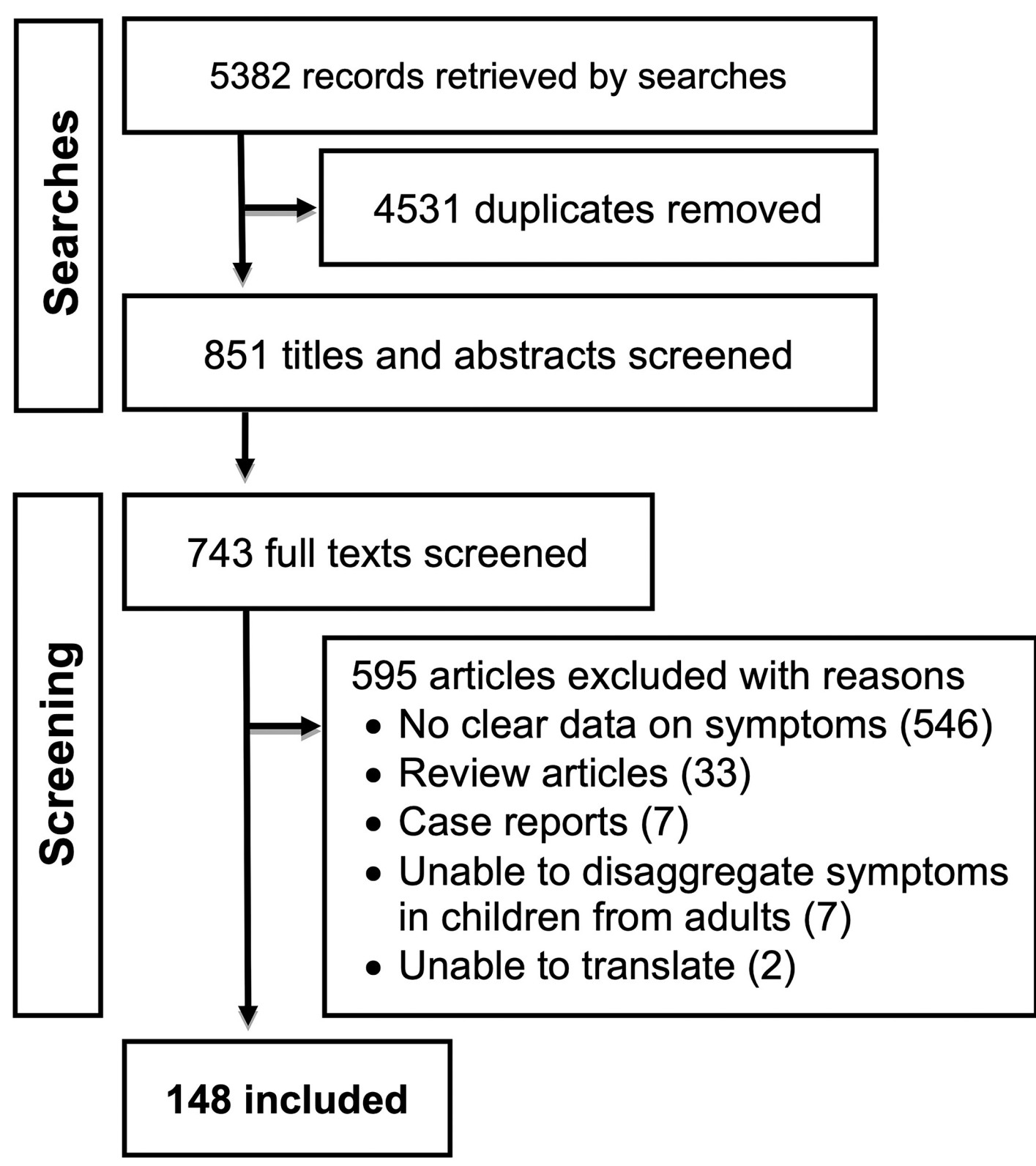

**Fig 1. Study flow chart.**

**Table 1. Meta-analysis of the prevalence of symptoms in adults with laboratory test confirmed COVID-19.**

| System | Symptom | Number of studies | Number of people | Prevalence (95% CI) | I² |
|---|---|---|---|---|---|
| Systemic | Fever | 138 | 21,701 | 78 (75, 81) | 94% |
| | Fatigue | 78 | 13,385 | 31 (27, 35) | 95% |
| | Myalgia | 72 | 11,389 | 17 (14, 19) | 88% |
| | Rigors | 17 | 2834 | 18 (13, 22) | 88% |
| | Arthralgia | 2 | 401 | 11 (8, 14) | / |
| | Rash | 1 | 1099 | 0 (0, 1) | / |
| Respiratory | Any cough (dry or productive) | 138 | 21,682 | 57 (54, 60) | 94% |
| | Dry (non-productive) cough | 136 | 17,380 | 58 (54, 61) | 93% |
| | Productive cough | 70 | 10,017 | 25 (22, 28) | 90% |
| | Dyspnoea | 94 | 12,713 | 23 (19, 28) | 97% |
| | Chest pain | 30 | 3510 | 7, (4, 10) | 92% |
| | Haemoptysis | 21 | 4698 | 2 (1, 2) | 42% |
| | Wheeze | 16 | 2013 | 17 (9, 26) | 96% |
| Ear, nose and throat | Sore throat | 78 | 11,721 | 12 (10, 14) | 88% |
| | Rhinorrhoea | 36 | 10,656 | 8 (5, 12) | 97% |
| | Vertigo / dizziness | 16 | 1972 | 11 (6, 16) | 90% |
| | Nasal congestion | 10 | 2584 | 5 (3, 7) | 78% |
| | Hyposmia | 3 | 317 | 25 (4, 55) | / |
| | Hypogeusia | 2 | 220 | 4 (1, 8) | / |
| | Otalgia | 1 | 68 | 4 (1, 11) | / |
| Gastrointestinal | Diarrhoea | 93 | 11,707 | 10 (8, 12) | 93% |
| | Nausea | 27 | 4584 | 6 (3, 10) | 95% |
| | Vomiting | 26 | 4959 | 4 (2, 8) | 94% |
| | Abdominal pain | 19 | 3331 | 4 (2, 7) | 88% |
| Central nervous system | Headache | 65 | 15,958 | 13 (10, 16) | 97% |
| | Confusion | 6 | 869 | 11 (7, 15) | 67% |
| | Ataxia | 1 | 214 | 0 (0, 2) | / |
| Eyes | Conjunctivitis | 9 | 2715 | 2 (1, 4) | 80% |
| | Ophthalmalgia | 1 | 534 | 4 (3, 6) | / |
| | Photophobia | 1 | 534 | 3 (2, 4) | / |

## Discussion

This review describes 24,410 adults with laboratory test confirmed COVID-19 from 9 countries. We confirm that the purported cardinal symptoms of fever and a new persistent cough are indeed the most prevalent symptoms of COVID-19 worldwide. However, we also show that at approximately 1 in 5 test-positive adults were never febrile and fewer than 3 in 5 developed a cough. Since the patients in the included studies are likely to have moderate-severe disease warranting hospitalisation and thus testing, it is likely that we over-estimate the true prevalence of symptoms in the population. Consequently, the use of symptoms alone to screening adults for SARS-CoV-2 infection is likely to miss a substantial number of infected individuals.

Our point estimates of the prevalence of fever (78%) and cough (57%) are approximately 10% lower than the estimates from prior reviews [8,10–13] which we feel might explained by two specific factors. Firstly, prior reviews [8,10–13] did not systematically search (or search at all [8,10,11,13]) for preprints uploaded to online repositories such as medRxiv or bioRxiv [15], both of which have seen a surge in uploads related to the COVID-19 pandemic [15]. This

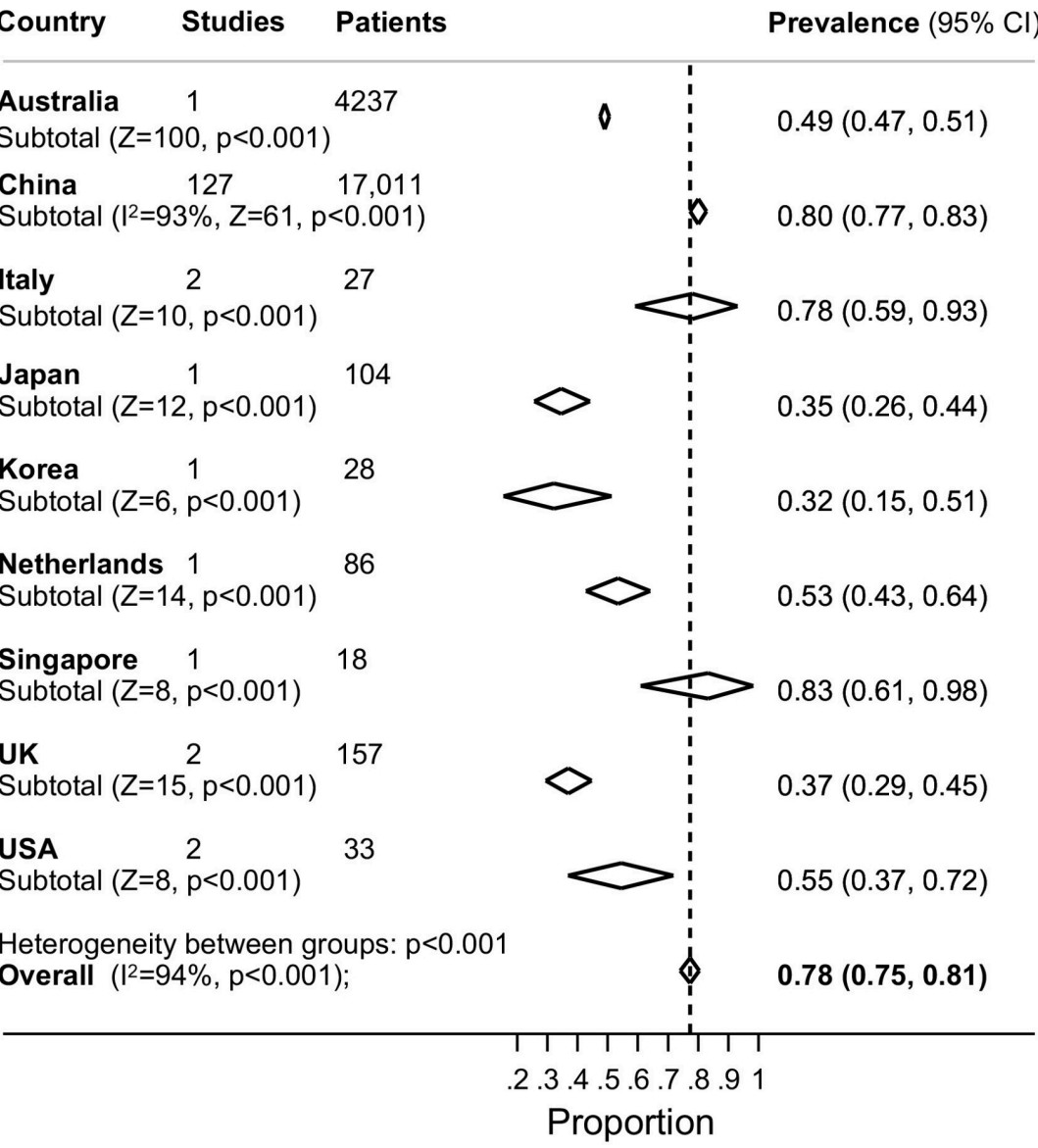

**Fig 2. Forest plot of the prevalence of fever in adults with laboratory test confirmed COVID-19.** The diamonds are summary estimates from each country.

explains why the largest and most recent other review (posted on March 25[th] 2020 in medRxiv [12]) included just 30 studies. Secondly, several weeks have passed since the other reviews [8,10–13] were performed and the delay from searching to posting a preprint in medRxiv was between 11 days [11] and 5 weeks [12,13,178]. For those articles not uploading a preprint, the delay from searching to publication was 3 weeks [10]. Therefore, it is likely that the prior reviews [8,10–13] had a higher proportion of adults with more severe disease (given that testing was limited to those admitted to hospital in the early phase of the outbreak) whereas more recent studies are likely to include adults with mild symptoms due to the wider availability of testing alongside natural progression of the disease. Conversely, more-recent studies of real-time population-wide tracking of self-reported symptoms in subsequently test-positive patients are essentially identical to our point estimates for cough [179]; however, data

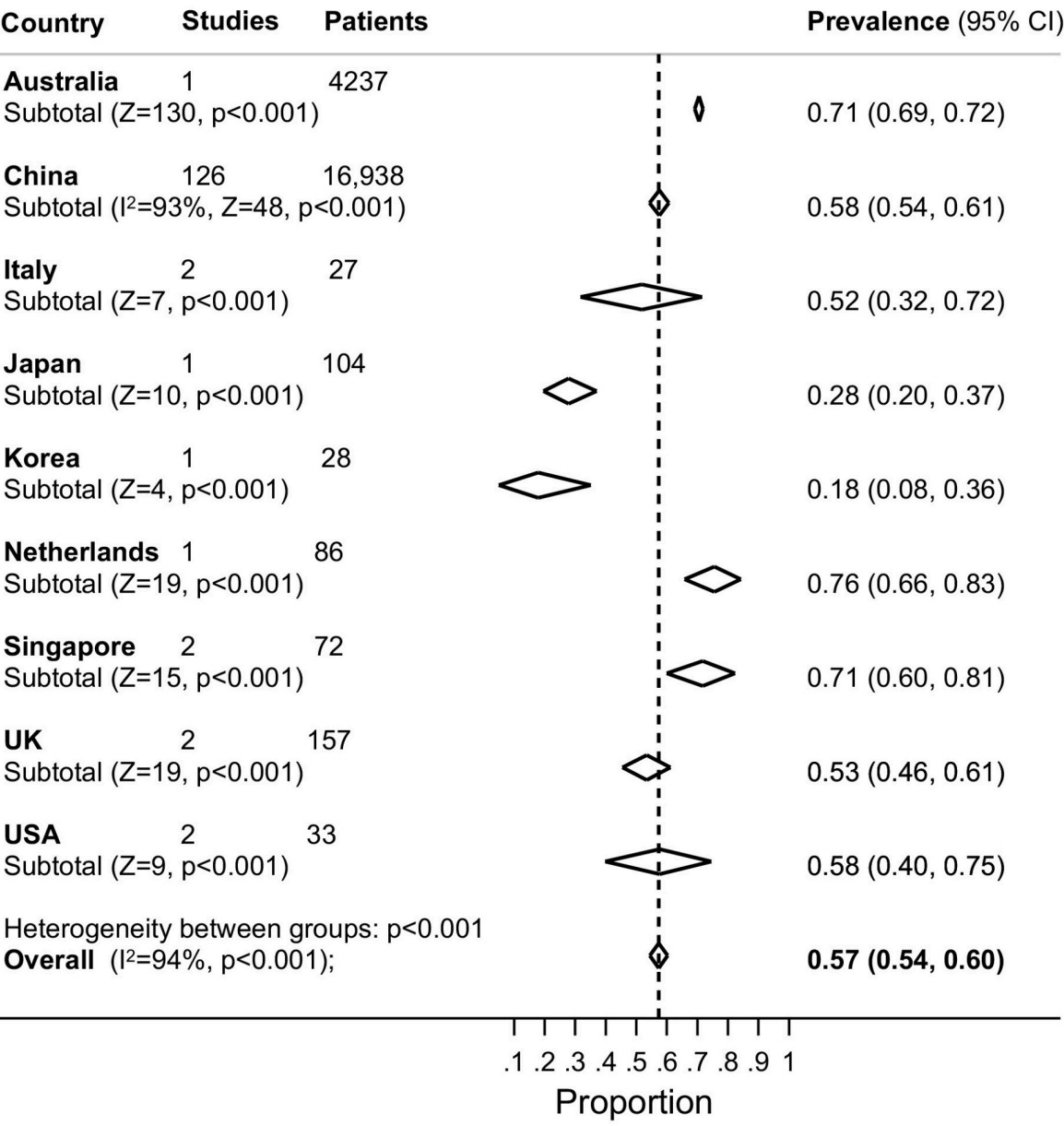

**Fig 3. Forest plot of the prevalence of cough (dry or productive) in adults with laboratory test confirmed COVID-19.** The diamonds are summary estimates from each country.

concerning fever [180] are less concordant but given the variability of core body temperature, methods of measurement and the definition of this symptom, variability is expected.

## Limitations

We acknowledge that there is both within-country and between-country differences in the estimated prevalence of different symptoms, which presents issues with regards to generalising the findings. However, the unique strength of a meta-analysis is the ability to compare datasets from difference sources, identify patterns and discrepancies [181], and no statistical technique is able to correct for weaknesses or idiosyncrasies of the original data. Differences in the study

designs, settings and what types of patients (mild, moderate or critically unwell) were sampled are all likely to be responsible for the observed heterogeneity. The sampling strategy is known to bias the prevalence of conditions and ideally, prevalence studies recruit a (non-probabilistic) consecutive sample because they are more likely to represent the target population. In comparison, convenience sampling (i.e. those with available data) and purposive sampling (e.g. reports of individuals with specific clinical features) which are common in the included studies, introduce selection bias and tend to upwardly bias estimates of symptom prevalence. Equally, enrolling patients from hospital settings rather than the community is more likely to upwardly bias the estimates of prevalence. Overall, we suspect that our results over-estimate the true prevalence of symptoms amongst test-positive adults.

In some instances, it was impossible to ascertain whether different publications which originated from the same hospital or region included (some of) the same subjects because the recruitment timeframes and sampling strategies were not sufficiently described in the study methods. We recommend that future publications detail (where possible) if their sample is also represented in other works and describe their methods in accordance with relevant reporting guidelines.

The way in which we extracted some of the data might bias the findings. We dichotomised fever (based on the definition in the parent study) and thresholds differed study-to-study (between 37°C and 38°C, S2 Table) which limits the transferability of the findings to clinical practice. We also did not extract data on combinations of symptoms (such as fever and cough together, or diarrhoea and vomiting for example) which was on oversight in the protocol development phase though equally, this was poorly reported in the literature. Future researchers who wish to build upon this dataset might consider extracting combinations of symptoms alongside isolated symptoms from the few studies were this is reported [174].

## Conclusions

We confirm that fever and cough remain the most prevalent symptoms of adults infected by SARS-CoV-2. However, there is a large proportion of infected adults which symptoms-alone do not identify. To expedite future iterations of this work, our data is freely available in the Open Science Framework repository.

## Supporting information

**S1 Checklist. PRISMA 2009 checklist.**
(DOC)

**S1 Appendix. Search strategy.**
(DOCX)

**S1 Fig.**
(DOCX)

**S2 Fig.**
(DOCX)

**S3 Fig.**
(DOCX)

**S4 Fig.**
(DOCX)

**S5 Fig.**
(DOCX)

**S6 Fig.**
(DOCX)

**S7 Fig.**
(DOCX)

**S8 Fig.**
(DOCX)

**S9 Fig.**
(DOCX)

**S10 Fig.**
(DOCX)

**S1 Table. Study characteristics.** See the bibliography at the end of the supplementary materials. RT = reverse transcriptase; PCR = polymerase chain reaction.
(DOCX)

**S2 Table. Meta-analyses of the prevalence of symptoms in adults with laboratory test confirmed COVID-19, subgrouped by country.**
(DOCX)

## Author Contributions

**Conceptualization:** Luke McGuinness, Ryckie G. Wade.

**Data curation:** Michael C. Grant, Luke Geoghegan, Zakaria Mohammed, Luke McGuinness, Emily L. Clarke, Ryckie G. Wade.

**Formal analysis:** Marc Arbyn, Luke McGuinness, Ryckie G. Wade.

**Investigation:** Marc Arbyn, Ryckie G. Wade.

**Methodology:** Michael C. Grant, Luke Geoghegan, Marc Arbyn, Zakaria Mohammed, Luke McGuinness, Emily L. Clarke, Ryckie G. Wade.

**Project administration:** Marc Arbyn, Luke McGuinness, Ryckie G. Wade.

**Resources:** Marc Arbyn, Ryckie G. Wade.

**Software:** Marc Arbyn, Luke McGuinness, Ryckie G. Wade.

**Supervision:** Marc Arbyn, Ryckie G. Wade.

**Validation:** Marc Arbyn, Ryckie G. Wade.

**Visualization:** Michael C. Grant, Marc Arbyn, Ryckie G. Wade.

**Writing – original draft:** Michael C. Grant, Luke Geoghegan, Marc Arbyn, Ryckie G. Wade.

**Writing – review & editing:** Michael C. Grant, Luke Geoghegan, Marc Arbyn, Zakaria Mohammed, Luke McGuinness, Emily L. Clarke, Ryckie G. Wade.

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
