## [Decision Letter · Decision Letter 0]

28 May 2020

PONE-D-20-13070

The prevalence of symptoms in 24,410 adults infected by the novel coronavirus (SARS-CoV-2; COVID-19): A systematic review and meta-analysis of 148 studies from 9 countries

PLOS ONE

Dear Dr. Wade,

Thank you for submitting your manuscript to PLOS ONE. After careful consideration, we feel that it has merit but does not fully meet PLOS ONE’s publication criteria as it currently stands. Therefore, we invite you to submit a revised version of the manuscript that addresses the points raised during the review process.

We believe that an update of searches is warranted prior to publication.

We look forward to receiving your revised manuscript.

Kind regards,

Jennifer A Hirst, DPhil

Academic Editor

PLOS ONE

Journal Requirements:

2. Please confirm that you have included all items recommended in the PRISMA checklist including a Supplemental file of the results of the quality assessment for each study assessed and an assessment of publication bias using graphical methods (e.g. Funnel plot) and statistical methods (e.g. Egger’s test) as appropriate.

'None'

Additional Editor Comments (if provided):

This is a well conducted piece of work worthy of publication.

However, this is a rapidly moving field and the reviewers feel that an update on searches is justified before publication.

Reviewers' comments:

Reviewer's Responses to Questions

**Comments to the Author**

1. Is the manuscript technically sound, and do the data support the conclusions?

Reviewer #1: Yes

Reviewer #2: Yes

2. Has the statistical analysis been performed appropriately and rigorously? 

Reviewer #1: Yes

Reviewer #2: Yes

3. Have the authors made all data underlying the findings in their manuscript fully available?

Reviewer #1: Yes

Reviewer #2: Yes

4. Is the manuscript presented in an intelligible fashion and written in standard English?

Reviewer #1: Yes

Reviewer #2: Yes

5. Review Comments to the Author

Reviewer #1: In this manuscript, Wade and collaborators present the results from a systematic review to identify the prevalence of COVID-19 symptoms worldwide. They pull together 148 studies from 9 countries for a total of 24,410 tested cases and find that fever and persistent cough are the most prevalent symptoms.

The study is well conducted, the authors screened several databases including pre-prints services. The statistical analysis is technically sound. However, as the authors correctly acknowledge, the COVID-19 research field is rapidly moving and my main concern is that the search was performed over a month and a half ago and hence may now be outdated. Indeed, very large community surveys (eg https://www.nature.com/articles/s41591-020-0857-9;
https://www.nature.com/articles/s41591-020-0916-2 among others) including over 3 mi people of which over 4500 with confirmed infection and over 140,000 with predicted infection have been published since. Given the difference in sample sizes the authors should include this as part of their quantitative analysis or at least mention that other studies have been conducted since April 5th and discuss whether their results/conclusions are still valid or not.

Reviewer #2: I recommend publication of this timely paper.

6. PLOS authors have the option to publish the peer review history of their article (what does this mean?). If published, this will include your full peer review and any attached files.

Reviewer #1: No

Reviewer #2: No

---

## [Editor Report · Decision Letter 1]

3 Jun 2020

The prevalence of symptoms in 24,410 adults infected by the novel coronavirus (SARS-CoV-2; COVID-19): A systematic review and meta-analysis of 148 studies from 9 countries

PONE-D-20-13070R1

Dear Dr. Wade,

We’re pleased to inform you that your manuscript has been judged scientifically suitable for publication and will be formally accepted for publication once it meets all outstanding technical requirements.

Kind regards,

Jennifer A Hirst, DPhil

Academic Editor

PLOS ONE
---

## [Editor Report · Acceptance letter]

5 Jun 2020

PONE-D-20-13070R1 

The prevalence of symptoms in 24,410 adults infected by the novel coronavirus (SARS-CoV-2; COVID-19): A systematic review and meta-analysis of 148 studies from 9 countries 

Dear Dr. Wade:

I'm pleased to inform you that your manuscript has been deemed suitable for publication in PLOS ONE. Congratulations! Your manuscript is now with our production department. 

Kind regards, 

on behalf of

Dr. Jennifer A Hirst 

Academic Editor

PLOS ONE